# Adapting Clofazimine for Treatment of Cutaneous Tuberculosis by Using Self-Double-Emulsifying Drug Delivery Systems

**DOI:** 10.3390/antibiotics11060806

**Published:** 2022-06-15

**Authors:** Daniélle van Staden, Richard K. Haynes, Joe M. Viljoen

**Affiliations:** Centre of Excellence for Pharmaceutical Sciences (Pharmacen™), Building G16, Faculty of Health Sciences, North-West University, Potchefstroom 2520, South Africa; dvanstaden711@gmail.com (D.v.S.); or haynes@ust.hk (R.K.H.)

**Keywords:** tuberculosis, multi-drug resistant TB, clofazimine, cutaneous tuberculosis, drug delivery, permeation enhancers, self-double-emulsifying drug delivery system, topical administration

## Abstract

Although chemotherapeutic treatment regimens are currently available, and considerable effort has been lavished on the development of new drugs for the treatment of tuberculosis (TB), the disease remains deeply intractable and widespread. This is due not only to the nature of the life cycle and extraordinarily disseminated habitat of the causative pathogen, principally *Mycobacterium tuberculosis* (*Mtb*), in humans and the multi-drug resistance of *Mtb* to current drugs, but especially also to the difficulty of enabling universal treatment of individuals, immunocompromised or otherwise, in widely differing socio-economic environments. For the purpose of globally eliminating TB by 2035, the World Health Organization (WHO) introduced the “End-TB” initiative by employing interventions focusing on high impact, integrated and patient-centered approaches, such as individualized therapy. However, the extraordinary shortfall in stipulated aims, for example in actual treatment and in TB preventative treatments during the period 2018–2022, latterly and greatly exacerbated by the COVID-19 pandemic, means that even greater pressure is now placed on enhancing our scientific understanding of the disease, repurposing or repositioning old drugs and developing new drugs as well as evolving innovative treatment methods. In the specific context of multidrug resistant *Mtb*, it is furthermore noted that the incidence of extra-pulmonary TB (EPTB) has significantly increased. This review focusses on the potential of utilizing self-double-emulsifying drug delivery systems (SDEDDSs) as topical drug delivery systems for the dermal route of administration to aid in treatment of cutaneous TB (CTB) and other mycobacterial infections as a prelude to evaluating related systems for more effective treatment of CTB and other mycobacterial infections at large. As a starting point, we consider here the possibility of adapting the highly lipophilic riminophenazine clofazimine, with its potential for treatment of multi-drug resistant TB, for this purpose. Additionally, recently reported synergism achieved by adding clofazimine to first-line TB regimens signifies the need to consider clofazimine. Thus, the biological effects and pharmacology of clofazimine are reviewed. The potential of plant-based oils acting as emulsifiers, skin penetration enhancers as well as these materials behaving as anti-microbial components for transporting the incorporated drug are also discussed.

## 1. Introduction

Control of the global tuberculosis (TB) burden has become a considerable concern in both developing as well as developed countries [1,2]. The countries with the highest TB incidence are India, Indonesia, Nigeria, the Philippines, Pakistan and South Africa [3]. It has been recently noted that there has been an increase in TB cases in Germany and Switzerland [4,5,6], a phenomenon that may be linked to the flow of African refugees to Europe [4,6,7,8]. In general, however, there has been an overall decrease of approximately 10% per annum in TB incidence in Europe since 2010. Nonetheless, it does appear that as a proportion of all individuals who have TB, Europe has the highest percentage of infections due to drug-resistant TB [9].

However, the drive to eliminate TB is not only limited to the aforementioned countries [3]. A recent publication describes the daunting challenge of maintaining control of TB in Australia [10]. Statistics indicate that 90% of TB cases reported in Australia during 2013 were of infected individuals that originated from other countries [11]. This occurrence is linked to the high mobility of modern populations, in this case, disease carriers coming in from neighboring countries such as Papua New Guinea [12]. On the other hand, the United States (US) experienced 22 years of declining TB incidence until 2015 when the number of TB cases increased for the first time since 1983. The incidence of multi-drug resistant TB (MDR-TB) in the US is described as stable, but special attention has to be paid to the vulnerability of infected children as MDR-TB affects both adults and minors alike [13].

Largely through difficulties in diagnoses and a generally active physical demeanor, TB in children is frequently misdiagnosed or overlooked. There is nonetheless a vast number of children infected with TB and MDR-TB, and more concerning is the fact that this number is increasing [14]. In TB endemic areas, it furthermore appears to be one of the ten main causes of death in children younger than 5 years of age [15]. Additionally, TB deaths in children are often misdiagnosed as being due to meningitis, pneumonia, malnutrition and/or HIV/Acquired Immunodeficiency Syndrome (AIDS). If these aspects are taken into consideration, deaths due to TB in children are considerably higher than as based on the initial correct diagnoses [14,16]. In addition, accurate estimates of the true MDR-TB burden among children are unavailable as TB displays a paucibacillary nature in children, which complicates confirmation of the disease [13]. Therefore, this disease is now receiving much closer attention in both developing and developed countries; this is mandatory for acquiring control over, and eventual elimination of TB [17,18].

For these reasons, it is recommended that the focus of TB treatment must shift to the global antimicrobial resistance crisis as less than 50% of patients treated for drug-resistant TB infections are efficiently cured. It is furthermore advocated that individualized therapy should be implemented [9,19]. Currently, poor adherence to anti-tubercular treatment regimens is attributed to prolonged treatment schedules and adverse reactions accompanied by extended administration [3]. In turn, poor adherence to first-line anti-tubercular regimens leads to the development of resistant TB isolates [3]. Therefore, the potential of including existing anti-tubercular agents, such as clofazimine, to first-line regimens might shorten treatment time and improve patient adherence [3].

The World Health Organization (WHO) implemented the “End-TB” campaign in September 2018 that had as its central aim the elimination of TB by the year 2035. To achieve this objective, three pillars of the WHO End TB Strategy were developed, where “intensified research and innovation” was included as one of the pillars. The two key components of this pillar are “discovery, development and rapid uptake of new tools, interventions and strategies” and “research to optimize implementation and impact, and promote innovations”. Moreover, at the first ever United Nations high level meeting on TB held in 2018, the commitment was made to increase research and innovation, with the development of new tools for TB treatment [20,21,22,23,24,25]. Therein, emphasis has now been placed on finding unique drug delivery systems, while utilizing “familiar drugs”, in addition to using more traditional approaches of discovering new and active compounds.

However, a drastic shortfall in aims of the End-TB Strategy has lately and starkly become evident, as is described in the WHO 2021 TB report [26]. The report clearly states that the global progress of the End-TB Strategy has been reversed to a 18% decline in newly diagnosed TB cases [26], thus representing a stark regression back to the situation as was noted in 2012 [26]. This drastic decline is attributed to the COVID-19 pandemic as the lock-down response greatly revoked the provision of essential TB services [26]. Therefore, it is clear that research into TB as outlined in the 2018 United Nations report must now be greatly intensified in order to now fulfil the aims of the WHO End-TB strategy. Therefore, this review aims to discuss the potential of adapting current anti-tubercular drug delivery by presenting clofazimine in Self-Double-Emulsifying drug delivery systems (SDEDDSs). Dermal drug delivery is suggested for the purpose of limiting adverse reactions associated with clofazimine while improving patient compliance as an additional therapy to provide topical treatment options for cutaneous TB (CTB). The possibility of plant-based oils acting as emulsifiers, skin penetration enhancers as well as these materials behaving as anti-microbial components for transporting the included drug(s) are also considered.

## 2. Anti-Tubercular Drug Resistance

TB is one of the principal epidemics that cause global morbidity and mortality. It is a fatal airborne infection, which is generated by *Mycobacterium tuberculosis (Mtb)* [26]. *Mtb* is categorized as an acid-fast bacillus with a slow growing rate [1]. Furthermore, this organism is able to survive in a quiescent state whilst within the human body without eliciting any TB symptoms [27]. This inactive TB state is referred to as latent TB infection (LTBI) [14,27]. Current research indicates that at least one in four people worldwide have latent TB. In almost 10% of the population infected with latent TB, this will develop into active TB. However, numerous individuals, including children, who may be living with TB patients in poor, overcrowded conditions, as well as immune-compromised patients, are especially vulnerable to LTBI. Subsequently, upon development of an immunocompromised situation such as through infection with HIV, development of diabetes mellitus, through an organ transplant, by becoming pregnant, or living in overcrowded and unhygienic environments coupled with poor nutritional intake, the inactive TB state may transform into an active TB infection. Persons, for example, infected with HIV, are 20 to 30 times more prone to contract active TB from LTBI [14,28,29,30,31].

In short, active TB is currently systemically treated and consists of two phases. The first being termed the intensive phase, which is about 8 weeks long. During this stage, first-line anti-tubercular agents namely, isoniazid, rifampicin, ethambutol and pyrazinamide, are concurrently administered. Following, once patients are no longer deemed infectious, they still require more treatment to successfully eliminate the infection, therefore, a maintenance phase is employed that entails concurrent administration of isoniazid and rifampicin. This stage is continued for at least 16–28 weeks, depending on the HIV status of a patient. It can even last up to 12 months. Importantly, patients need to adhere strictly to this treatment regimen to avoid drug resistance [32,33,34].

What is more concerning however is that the WHO indicated that 458,000 of the 10 million active TB cases reported in 2017 were triggered by *Mtb* with specific resistance towards rifampicin and isoniazid, which are two of the four first-line drugs normally used against active TB [25,26,35]. MDR-TB is defined as resistance to at least two of the front line anti-tubercular agents and is more prevalent in patients not adhering to anti-TB therapy. Multi-drug resistance is furthermore predictive of a poor outcome since these drugs are the key constituents of anti-TB treatment [25,35,36,37,38,39,40]. Estimates from the WHO 2021 TB report indicated a decline in the number or patients that were enrolled for treatment of MDR-TB from 2018 to 2020. However, this is by no means a good report as it only indicated that fewer patients registered for MDR-TB treatment and adhered to the treatment guidelines. This was probably due to the global COVID-19 pandemic lock-down responses, which in turn, significantly prevented essential TB services from being performed [24]. Thus, this extremely infectious disease is rendered even more complex due to the increasing occurrence of antimicrobial drug resistance.

Nevertheless, how does drug resistance occur given that TB is both preventable and treatable? As stated, TB drug resistance principally means that *Mtb* is resistant to first line anti-tubercular agents including isoniazid, rifampicin, pyrazinamide, streptomycin and ethambutol [36,41]. In order for a TB regimen to be effective, the simultaneous administration of various bactericidal and sterilizing anti-tubercular agents must be maintained for a sufficient period of time [1,42] in order to guarantee anti-microbial effectiveness together with the prevention of *Mtb* mutations that might lead to drug resistance [40]. Resistance generally emerges because of inadequate TB management when anti-tubercular drugs are misused or mismanaged. Examples include:patient noncompliance (failure to complete the correct full course of treatment),reinfection with *Mtb* post treatment,the inappropriate utilization of medicines such as when health-care providers prescribe the incorrect treatment regime (incorrect employment of antibiotics),prescribing an incorrect dosage, or length of time for treatment (incorrect treatment regime),an inconsistent treatment (drug availability),restricted access to treatment, orwhen the requisite drugs to be used are of poor quality [31,43,44].

It has even been found that clinical and genetic risk factors may also encourage the development of MDR-TB in non-HIV infected patients [45].

Even so, it has been established that the risk of development of drug-resistant TB is higher among HIV-infected individuals compared to HIV-seronegative patients. A study conducted in 1993–1994 in the USA found that HIV-infected patients with TB who had not previously been treated for TB were already infected by bacterial isolates with an 11.3% incidence of isoniazid resistance and an 8.9% incidence frequency of rifampicin resistance. These numbers were almost double those recorded from an HIV-negative population [46]. Although the mechanisms involved in the development of rifampicin resistance are not yet clearly understood, the poor absorption of rifampicin and ethambutol in HIV-infected patients may be associated with acquired drug resistance leading to treatment failure [45]. As portrayed in Figure 1, *Mtb* is combated with multiple anti-tubercular agents attacking this deadly infection with different mechanisms of action.

MDR-TB is generally treated with fluoroquinolones (moxifloxacin and ofloxacin) and injectable agents (amikacin, capreomycin, kanamycin or streptomycin), normally employed as last line anti-tubercular agents, in combination with p-aminosalicylic acid, cycloserine or ethionamide for at least 18 months [37,44]. Table 1 indicates the latest classification system provided by the WHO to initiate a prolonged, but effective MDR-TB treatment regimen. Clearly, this represents a complex treatment regimen that is costly and significantly extended, which furthermore tends to enhance lack of compliance. Moreover, in 8.5% of all patients that contracted MDR-TB, *Mtb* is also found to be resistant to at least one fluoroquinolone and/or one of the second-line injectable aminoglycosides. Resistance to both fluoroquinolones and aminoglycosides is described as extensively drug resistant TB (XDR-TB), whereas pre-XDR refers to resistance either to fluoroquinolones or aminoglycosides, respectively [25,35,40,48,49,50].

The emergence and transmission of these types of TB are again partly driven by the association between TB and HIV (TB/HIV coinfection). Evidence shows that populations infected with HIV are more prone to contract XDR-TB, which is fundamentally untreatable with normal TB treatment regimens. XDR-TB is a critical health hazard, predominantly in communities with a high incidence of HIV [44]. According to the 2018 WHO global TB statement, treatment is only effective in 55% of individuals that contracted MDR-TB, and is only effective in 34% of XDR-TB patients [25,35]. The presence of MDR-TB and XDR-TB furthermore enhance the occurrence of extra-pulmonary TB infections [52]. Therefore, treatment is even more complicated as 5% of all new TB cases are considered multi-drug resistant, with the global estimate being that 3.3% of first time TB cases and 20.5% of previously treated TB patients display MDR-TB [32,38].

## 3. Cutaneous Tuberculosis, an Extra-Pulmonary Infection

TB presents normally as a pulmonary infection. However, approximately 20% of TB cases are extrapulmonary (EPTB) and this number is rising; consequently, it cannot be ignored any longer. EPTB includes ocular TB, hepatobiliary and splenic TB, TB spondylodiscitis, pleural TB, TB encephalitis, breast TB, TB pericarditis, TB lymphadenitis, gastrointestinal and peritoneal TB, genitourinary TB, osteoarticular TB, TB meningitis and cutaneous TB (CTB).

The incidence of EPTB increases with associated HIV infection, an increase in MDR-TB and immunosuppressive treatment, specifically utilizing inhibitors of tumor necrosis factor-alpha TNFα [1,32,33,53,54,55,56,57,58,59,60,61,62].

CTB is a relatively uncommon disease where merely 1–2% of all EPTB infections involve cutaneous association and of these cases only about 0.1–1% of all cutaneous disorders can be related to CTB [32,54,59,63]. CTB was first described in 1826 by Théophile Laennec, inventor of the stethoscope, who reported a lesion on his hand that was evidently triggered through mycobacterial penetration through his skin. The causative organism was however only identified in 1882 when Robert Koch first identified the bacterium *Mtb* as the TB pathogen [32,33,54,55,59,63,64,65].

Skin manifestations due to EPTB are normally caused by *Mtb* and are known as true CTB. However, other atypical *Mycobacterium* species may also account for cutaneous manifestations. These Mycobacteria can be divided into two sub-categories, namely: slow growers (*M. haemophilum*, *M. leprae*, *M. marinum*, *M. ulcerans*, and less regularly the *M. avium* complex, *M. kansasii*, *M. malmoense*, *M. scrofulaceum*) and rapid/fast growers (*M. chelonae*, *M. fortuitum*, and infrequently *M. abscessus*). Rapidly growing organisms have a 7 day or less incubation period, whereas slow growers have a longer incubation time [32,66,67,68,69,70,71]. *M. bovis* and the Bacille Calmette-Guérin vaccine, an attenuated strain of *M. bovis*, have moreover been reported to cause CTB [70]. Infections by means of atypical mycobacteria appear primarily in immunocompromised populations and there is a tendency for dissemination, whereas CTB infection in immunocompetent hosts follows skin penetration. This type of CTB is normally described as localized [55].

True CTB is more prevalent among women and young adults. The most frequently affected area of the skin is the face, although it commonly appears on the neck and torso as well [71]. Factors contributing to the diverse manifestations of the different CTB sub-categories include age and sex. For example, scrofuloderma is more common amongst pediatric patients, whereas erythema induratum of Bazin presents more frequently in females, and TB verrucosa cutis is predominantly observed in males [72,73,74]. The incidence of divergent CTB manifestation is displayed in Figure 2.

Individuals exposed to malnutrition or unhygienic conditions, or suffering from diabetes mellitus, end-stage renal disease, malignancies, HIV co-infection, or through submission to immunosuppressive therapy, and infants and pregnant women, amongst others, are at risk of contracting active CTB [26,76,77]. Overall, a diverse collection of individuals is vulnerable, especially due to the global increase in MDR-TB and XDR-TB [78].

Currently three criteria are used to classify CTB, namely, pathogenesis of the causative organism, clinical presentation, and histology. CTB is therefore generally categorized as either an endogenous or exogenous propagation, considering the bacterial basis and the infection means, as well as a multibacillary or paucibacillary configuration, depending on the bacterial load in the lesion. The exogenous mechanism refers to direct inoculation into the skin, whereas the endogenous mechanism is secondary to a previous infection where transmission arises by means of contiguous, lymphatic or hematogenous dissemination [79,80,81,82,83]. This variety of clinical CTB manifestations severely complicates diagnoses. Moreover, CTB is an elusive dermal condition as it resembles a number of visually similar dermal disorders [32,63,76,79]. In addition, the microbial basis of CTB infection (Figure 3) may be difficult to define, even though improved detection techniques including culture smears, polymerase chain reaction examinations and enzyme-linked-immunosorbent serologic assays are available [33,34,63,79,83,84].

Currently, treatment options for CTB remain the same as for general systemic TB due to the involvement of systemic infections in most cases. Furthermore, surgical interventions involving excision of lesions and correction of deformities in certain indications are sometimes necessary [32,55,85]. CTB treatment also comprises two phases. The first phase is the intensive phase and lasts approximately 8 weeks. This phase consists of simultaneous administration of first line anti-tubercular agents. Following this regimen, patients are no longer considered infectious, but do require more treatment in order to eradicate the infection. Subsequently, a maintenance phase is implemented consisting of simultaneous administration of isoniazid and rifampicin. This phase is sustained for a period of at least 16 weeks for HIV negative patients and 28 weeks for HIV positive patients, but nonetheless may even last for 9–12 months. For the treatment to be successful, patients need to adhere stringently to this treatment regimen [32,33,34]. However, due to these long-term treatment phases, patient compliance may be problematic, and in order not only to improve patient compliance, but also enhance positive treatment outcomes, alternative solutions are required. Firstly, anti-microbial agents other than conventional TB drugs must be explored in order to shorten the treatments required to cure MDR-TB and XDR-TB, and to minimize the adverse effects that are normally associated with first-line anti-tubercular compounds. Likewise, novel therapeutic combinations of existing antibiotics and first line TB agents may be more beneficial as recommended by the WHO and United Nations [2,19,20,21,22,23].

Recently, interest in so-called “orphan drugs”, i.e., drugs intended for rare diseases that for example affect less than 200,000 people per annum in the USA, has increased despite the fact that these drugs are considered orphans both therapeutically and commercially due to the minor patient population in need of these compounds [80]. Interestingly, orphan drugs are attracting attention from researchers aiming to treat dermal conditions since many side effects can be circumvented by utilizing dermal applications [80].

Clofazimine is one such compound that notably displayed a significant efficacy against resistant TB strains in the ‘Bangladesh regimen’ described below [2,25,81]. Individualized anti-tubercular therapy is furthermore increasingly recommended, as it is seen as the most effective and safe manner of targeting TB resistance in individual patients without inducing further resistance [82,83]. In order to turn the tide against TB and CTB, reintroduction of agents such as clofazimine should not be disregarded as these can assist in both individualized therapy where drugs are only employed if effective against the infection as indicated by predictive tests, as well as contributing to shortening of the treatment phases [2,48,84].

## 4. Clofazimine, an Orphan Drug

Clofazimine, initially known as “B663”, was originally synthesized in 1954 by Vincent Barry and co-workers to treat TB [85]. It is a highly lipophilic riminophenazine dye, which is significantly active against *Mtb*. However, despite high bactericidal and sterilizing activity in vitro as well as in a mouse model of MDR-TB, and its good oral absorption in micronized form along with its deposition and persistence in adipose tissue and in cells of the reticuloendothelial system, this compound did not initially progress as an anti-tubercular agent as it was found to be ineffective against TB in humans [14,86,87,88,89,90,91]. Nonetheless, it was concluded that due to the accumulation of clofazimine inside macrophages, it should be effective against intracellular diseases [92]. Efficacy against leprosy was noted at the end of 1959, and after clinical trials during the 1960s in Nigeria and elsewhere, the Swiss pharmaceutical company Novartis marketed clofazimine in 1969 under the brand name Lamprene^®^ as an anti-leprosy agent [25,93,94].

In 1982, a WHO Study Group recommended utilizing clofazimine as a constituent of a triple drug combination with dapsone and rifampicin for treatment of multibacillary leprosy [88,94]. However, Novartis was only granted FDA approval of clofazimine at the end of 1986 and its use as anti-tubercular agent lapsed for several decades [93,95,96]. Moreover, clofazimine was classified as a Group 5 drug by the WHO, indicating that it is a drug with unclear efficacy that is not recommended for routine use [97].

Later, interest in clofazimine for the treatment of TB was awakened following its inclusion in the so-called “Bangladesh regimen” where it was established that approximately 90% of patients that contracted MDR-TB were healed within a 9–11 month period [2,81,98]. This was a substantial improvement over the standard WHO-proposed 24 months regimen that cured less than 50% of MDR-TB patients. Results from a larger clinical trial study in Bangladesh and from other clinical trials in Cameroon and Niger confirmed these conclusions [89,99,100,101]. The Bangladesh regimen according to the WHO 2016 guidelines comprises an initial treatment phase consisting of 4–6 months’ treatment with kanamycin, moxifloxacin, prothionamide, clofazimine, pyrazinamide, isoniazid (in high dosages) and ethambutol. This initial phase is followed by a 5 months treatment period with moxifloxacin, clofazimine, pyrazinamide and ethambutol [2,100]. The specific bactericidal and treatment-shortening action of clofazimine was furthermore noted in MDR-TB mouse models, and it was likewise apparent in a controlled clinical trial performed in China [102,103,104].

In contrast, the randomized controlled STREAM trial indicated non-inferiority of the short-course-containing regimen, but not superiority [105]. For these reasons, the full potential of clofazimine as an anti-tubercular agent has not yet been established. Nonetheless, clofazimine is listed in the WHO Model List of Essential Medicines as a leprostatic-, anti-tubercular- and anti-inflammatory agent [106]. It is also categorized by the WHO as a Group C agent, that is, other core second line anti-tubercular agents [48]. Therefore, clofazimine may have the capacity to significantly advance the treatment of MDR-TB and CTB.

Oral absorption of clofazimine is variable, fluctuating from 45–62% when taken on an empty stomach; however, absorption is increased when ingested with food. This drug is metabolized by glucuronidation in the liver, incompletely expelled by the biliary route, and only negligibly excreted in urine, sputum, sebum, and sweat. Clofazimine has been reported to cross the placenta and is excreted in breast milk [90,95,107,108]. Second line anti-tubercular drugs are known to be more toxic [109] and clofazimine is no exception as the highly lipophilic nature of this drug leads to accumulation in macrophages [92]. However, it must be noted that human macrophages are the primary host cells for *Mtb*, and thus, such a property will be of decisive clinical advantage when using clofazimine [92]. Moreover, accumulation of clofazimine in adipose cells such as in the heart, liver, breasts, adrenal glands, pancreas, spleen, bone marrow and lamina propria of the jejunum, extends the half-life to 65–70 days [88,89,90,95,107,108,110,111,112,113,114]. However, such accumulation leads to adverse reactions including reversible skin, hair and corneal dyschromia to brownish-black, dark red, or even to orange pink. Approximately 46% of patients receiving clofazimine treatment display conjunctival deposition, whereas nearly 53% experience crystal deposition in the cornea and 32% of patients present with clofazimine crystals in their tears [114,115]. Deposition of a dark-brown pigmentation is furthermore sometimes visible on the fingernails. Discoloration is more evident in fair-skinned individuals exposed to sunlight and even though it is reversible, it may take years to be completely resolved. Breast milk, sweat, urine, and feces may also discolor. In addition, dry skin and ichthyosis, defined as an increase in skin thickness due to lack of shedding of dead skin cells, often transpires. Severe gastrointestinal reactions (nausea, vomiting, abdominal pain, diarrhea) are rather common (reported in >10% of patients) and deemed most concerning, as accumulation, as well as precipitation of clofazimine in the wall of the small intestine, ensue after sustained administration [88,95,102,107,108,110,111,116,117]. In 1–10% of patients, weight loss, pruritus, diminished visual perception, and/or eye irritation/dryness occur. Less commonly, more severe adverse effects may be observed, including enteropathy possibly complicated by intestinal obstruction, gastrointestinal bleeding, and splenic infarction [107,108,118,119].

Normally, side effects accompanied by clofazimine treatment are dependent on the dose administered and the duration of administration [87]. Clofazimine up to 300 mg daily is often well tolerated; however, 100 mg per day is normally recommended due to gastrointestinal intolerance. Higher dosage regimens given over longer periods, i.e., beyond 4 months, may lead to crystal enteropathy and malabsorption, which is accompanied by effusive vague and/or severe abdominal distress. Coincidentally, discontinuation of treatment normally leads to a reasonably quick clearing of symptoms. Granting the fact that gastrointestinal side effects are not infrequent and have been observed in roughly one third of patients treated with clofazimine, the severity is usually not substantial enough to evoke termination of the treatment [14,95,102,107,108,113,118,119,120].

Fortunately, clofazimine is predominantly well-tolerated in children as observed from pediatric leprosy treatment [14]. It is classified as Pregnancy Category C, though little research has been conducted regarding its use to treat MDR-TB during pregnancy and limited data exist for later life teratogenicity. From data obtained from case reports, it was found that only 5 MDR-TB pregnant patients received clofazimine as part of the treatment regimen. All infants born were considered normal. Moreover, no significant adverse reactions due to clofazimine as part of the leprosy treatment regimen during pregnancy have been reported [88,107,108,121,122]. Clofazimine is furthermore safe to co-administer with antiretroviral therapy [123] and no known significant drug-drug interactions have been reported [88,121]. Thus, overall, insight into the physicochemical characteristics of clofazimine enables the design of novel drug delivery systems, and it should be possible to circumvent the most serious adverse effects of severe gastrointestinal reactions and accumulation of clofazimine in the gastro-intestinal tract. Clearly, the topical administration route will be advantageous as gastrointestinal accumulation can be bypassed and patient compliance not compromised [124]. Thus, an alternative route of administration together with a unique clofazimine dosage form will fortify this orphan drug with a more user-friendly profile.

## 5. An Alternative Strategy to Deliver Clofazimine More Effectively

### 5.1. Contemplating Topical Delivery

As stated, oral delivery of clofazimine is not only erratic, but also evokes numerous systemic adverse effects. On the other hand, topical drug delivery represents “the application of a drug onto the surface of the body to a localized area, such as the nose, eyes, rectum, vagina or skin, in order to establish relief of a condition restricted at the area of application, without creating any systemic effect” [125,126].

Over the past three decades, the trend to deliver drugs via the skin has been markedly increasing. It is becoming a considerably more popular alternative administration route, especially when aiming to avoid complications that accompany oral drug delivery [124]. Through implementation of this route to treat various topical ailments, patient compliance is increased, the risk of drug and food interactions is decreased, and hepatic metabolism is circumvented amongst others [127,128], thus rendering the dermal application route conspicuously beneficial [129]. However, the skin is designed to fulfil a barrier function, and subsequently permeation of various drugs, in concentrations that are of clinical relevance, is limited [130,131]. The barrier function of the skin is linked to the lipophilic nature of the outermost layer of the skin, namely the stratum corneum (SC) [131]. Therefore, the main challenge when formulating a topical drug delivery system is to attain an optimum concentration of the encapsulated drug at the site of action for a sufficient period of time [125,132].

The skin can be described as a multi-layered organ [133]. Each layer fulfils its own role in protecting the body, maintaining hydration and allowing sufficient blood perfusion to the skin [134,135,136]. The skin consists of three layers, namely the epidermis, dermis and hypodermis [137]. Epidermis thickness varies according to the cell layer thickness at the different skin regions of the body [135]. This layer is divided into two distinct fragments, the SC and viable epidermis [134]. The SC forms the outermost layer of the epidermis and is the lipophilic, protective layer of the skin that complicates drug delivery of a vast range of substances [136]. This protecting function of the SC is enforced by its lipophilic nature [134]. The SC consists of lifeless, anucleated cells comprising corneocytes linked by corneodesmosomes surrounded by a lipid matrix containing a mixture of cholesterol, triglycerides, free fatty acids and ceramides [134]. The second skin layer, the dermis is the most biochemically active layer of the skin [134]. It is responsible for body temperature regulation, cutaneous sensation, skin elasticity, blood circulation in addition to skin nutrition, and the removal of toxic substances [134,135]. Below the dermis is the third skin layer known as the hypodermis [137]. Subcutaneous fat is encased within the hypodermis [135,138]. Thereby, it acts as a supportive membrane for both the dermis and epidermis [135,139]. In order for a drug to be delivered through the skin and into the systemic circulation, it must be able to move through the lipophilic SC and the underlying hydrophilic viable epidermis and dermis, respectively. Consequently, a drug must have both hydrophilic and lipophilic properties in order to facilitate successful transdermal drug delivery [134,136].

There are three pathways that enable drug diffusion through the skin to occur [138]. First, a drug enters the skin through the intracellular or intercellular pathway [137]. Highly hydrated keratins are part of the corneocytes that are the main building blocks of the epidermis, where some components are able to traverse these cells. Moreover, hydrophilic drugs predominantly pass through the skin via the intracellular pathway [134]. Secondly, intercellular permeation may occur via diffusion of a compound through the lipid matrix of the skin as mostly observed for lipophilic drugs [134,137]. The third route, the transappendageal pathway, is restricted as a mere 0.1% of the skin surface is occupied by hair follicles and sweat glands that present an opportunity for modest transappendageal permeation of drugs [136,137]. Here, an additional limiting step for permeation of hydrophilic composites is the lipid-rich nature of the sebum that fills the sebaceous glands [140]. On the other hand, however, delivery of lipophilic drugs may be enhanced through this route [134]. Sweat, on the other hand, provides a pathway for the delivery of hydrophilic drugs via the transappendageal pathway [137], although drug delivery in the presence of sweat is limited as diffusion must occur against the diffusion gradient [140]. Hence, the physicochemical properties of a drug are highly important for the development of a drug delivery system that may enable topical and/or transdermal drug delivery [141]. The skin anatomy and dermal drug penetration pathways are displayed in Figure 4.

### 5.2. Clofazimine Characteristics Challenging the Topical Route

The physicochemical characteristics of clofazimine are presented in Table 2. Molecules with molecular weights less than 500 Da, such as clofazimine, with a molecular weight of 473.40 Da [142], are capable of relatively facile diffusion through the skin [141,143]. As skin is a non-homogenous medium, smaller particles will display increased skin permeation [144]. Clofazimine furthermore exhibits an exceptionally long half-life, as noted [94], which is beneficial since an extended half-life may require less frequent dosing. In turn, this characteristic might even reduce the adverse skin discoloration normally associated with the oral absorption of clofazimine [145].

Generally, the rate of diffusion through the skin will only occur at a rapid transfer rate if the concentration of clofazimine present on the skin surface is saturated [145]. It has been established that supersaturation on the skin can increase skin penetration and permeation of lipophilic drugs [147]. The diffusion and partition parameters assumed for lipophilic compounds indicate that supersaturation relates to increased thermodynamic activity without apparent effect on the barrier function of the skin. Hence, the clofazimine concentration incorporated into formulations, as well as its solubility in these formulations will significantly contribute toward improving the transfer rate through the skin [146]. Elsewhere, it has been established that the melting point of a compound may be used for evaluation of compound purity, as well as being used as a predictor of compound solubility [148]. In general, a melting point of less than 200 °C is recommended to ensure adequate solubility. Clofazimine has a slightly higher melting point (210–212 °C) than recommended, and although this difference is only marginal, drug solubility might possibly present some challenges [94,141]. The solubility parameter of a drug destined for dermal delivery should be similar to that of the skin (approximately 10 cal/cm^3^) [149]. An aqueous solubility of more than 1 mg/mL is deemed most favorable for a drug that needs to diffuse through the skin [94].

Clofazimine possesses a Log P value of 7.66, reflecting its highly lipophilic nature [150,151]. The Log P value is the n-octanol-water partition coefficient, which will reflect the partitioning of a drug between the lipophilic SC and the hydrophilic viable epidermis and dermis [152]. Thus, the high Log P value of clofazimine negatively affects skin permeation as the ideal Log P range, identified for topically delivered drugs, is approximately between 1 and 3 [142]. Overall, the high lipophilicity and very low aqueous solubility add to the challenges of formulating clofazimine into a topical/transdermal drug delivery system [95].

Another property to consider is the pH of the skin where the SC is able to tolerate a pH range from 5–9 [141]. For this reason, the pH of a dermal formulation is an important factor that must be considered in order to avoid skin irritation [153]. Moreover, the pH of a formulation will determine the fraction of unionized molecules available at the skin surface [154]. Drug molecules presented at the skin surface in an ionic form will cross the SC via the transappendageal route [155]. Conversely, unionized drug molecules will enter the skin via the intercellular (lipid) route [156]. The acidic or basic properties of a drug can determine which physicochemical properties are more important during topical or transdermal drug delivery [157]. For weakly acidic drugs, the order of importance is molecular weight > Log P > solubility; whereas the order of importance for weakly basic drugs, such as clofazimine is molecular weight > solubility > Log P.

Overall, although the effective topical delivery of highly hydrophobic compounds for example clofazimine faces various challenges, these may be overcome by utilizing excipients as well as unconventional drug delivery systems. For example, natural oils initiate reversible fluidization and rearrangement of the lipid matrix within the SC that facilitates penetration enhancement of drugs. Thus, the inclusion of natural oils may assist topical drug delivery [158]. In addition, natural oils are frequently employed during the formulation of dermal drug delivery systems as stabilizers, solubility enhancers and emulsifiers. Furthermore, the application of natural oils onto the skin is considered safe and generally acceptable to the public [159,160].

### 5.3. Auxiliary Natural Excipients to Enhance Topical Clofazimine Delivery

As stated by the WHO [160], approximately 80% of communities located in developing countries deploy ethno therapy, including topical application of natural oils as a more affordable health care and cosmetic alternative where skin pathologies are problematic. Insights provided by current research indicate that the topical application of natural oils may be accompanied by mutually beneficial properties influenced by the composition of the individual natural oil together with its method of extraction from usually a plant, or less commonly, an animal source. These properties may include enhanced skin barrier homeostasis, anti-oxidative activity, anti-inflammatory action, anti-microbial properties, promotion of wound healing, and possible anti-carcinogenic characteristics [161,162,163,164,165,166].

Unrefined plant-based oils, cold-pressed or cold extracted, are not exposed to heat or chemical processes during manufacturing, and thus the components remain unaltered [166,167,168,169,170]. Naturally occurring sediments, such as wax, in unrefined plant-based oils may improve dermal occlusive effects that inhibit skin dehydration whilst providing prolonged contact time with the skin to improve drug delivery. Unrefined natural oils retain the highest nutrient concentration together with unaltered fatty acids, and display a decreased tendency towards skin irritation [163,169,170]. On the other hand, cutaneous lipid disruption induced by fatty acids present within natural oils is required for skin penetration for delivery of the drug [167]. Thus, the extent of skin penetration may vary due to the unique composition of the plant oil selected in a topical formulation [158]. Further, the amount of drug which dissolves will be dependent on the nature of the oil, that is, different concentrations of dissolved drug are obtained for different oils, as has been observed for ibuprofen [171].

Free fatty acids are released when natural oils are metabolized within the skin [158]. They are of considerable importance for dermal drug delivery. The fatty acids present in plant-based oils disrupt the lipid structure of the SC enabling entry of the drug into the skin. The chemical structure, degree of unsaturation of the fatty acid and alkyl chain length influence the extent of skin penetration [172]. Interestingly, it has been found that unsaturated fatty acids have improved cutaneous penetration compared to saturated fatty acids with identical alkyl chain lengths [172]. This is attributed to improved capacity to disrupt the dermal lipid structure due to the more rigid chain structure of unsaturated fatty acids [172]. Not unexpectedly, combining different plant-based oils can provide a synergistic enhancement of dermal penetration [172].

Fatty acids can also contribute towards the barrier repair function of the skin. The best-known fatty acid for effecting skin penetration enhancement by means of disrupting the lipid structure of the SC is the essential mono-unsaturated fatty acid oleic acid. In contrast, the essential di-unsaturated fatty acid linoleic acid repairs the barrier function of the skin as it restores the lamellar phase of the lipid matrix within the SC [158,162,163]. In general, oleic and linoleic acids predominate in individual plant-based oils [163]. However, the optimum ratio of oleic acid to linoleic acid for favoring dermal drug delivery is still unknown [163].

Overall, designing and optimising dermal drug delivery systems for drugs as lipophilic as clofazimine present unique challenges. We next consider the potential of SDEDDSs compared to other lipid-based dermal drug delivery vehicles for topical delivery of clofazimine.

## 6. The Prospect of Self-Double-Emulsifying Drug Delivery Systems for Enhancing Topical Delivery of Clofazimine

### 6.1. Lipid-Based Carrier Systems Ideal for Lipophilic Drugs

It is estimated that approximately 30% of current commercial drugs including clofazimine and up to 50% of newly discovered drugs are highly hydrophobic [152,153]. Thus, lipid-based carrier systems are receiving special attention in the pharmaceutical industry in order to improve delivery of these drugs, especially those displaying poor aqueous solubility [173,174,175,176]. Encapsulation of the lipophilic drugs into inert lipid vehicles is shown to provide reproducible drug concentration profiles [173,177]. Lipid carrier systems include oils, surfactant dispersions, emulsions, nanoemulsions, liposomes, solid lipid carrier systems, nanostructured lipid carrier systems, self-emulsifying drug delivery systems (SEDDSs) and SDEDDSs [173,177,178,179]. Overall, lipid carrier systems can facilitate targeted drug delivery, thereby improving therapeutic effects and decreasing side effects, which will be important for anti-tubercular treatment [178].

### 6.2. Why SDEDDSs Are Considered More Appropriate Than Other Lipid-Based Carrier Systems

Liposomes, emulsions and nanoemulsions are frequently employed as drug delivery systems. These lipid-based formulations are also attracting increasing attention for enhancing topical and transdermal drug delivery [178,179,180,181]. Liposomes are spherical, bilayer structures of 100–200 nm in diameter. They are generally prepared from phospholipids and are able to encapsulate a wide range of diverse drugs, either in their aqueous central compartment or within the lipid bilayer. Their size, number of bilayers and various other components that may be included bestow their distinct characteristics. For example, inclusion of a surfactant results in the formation of transferosomes that are able to cross the SC. Inclusion of ethanol produces more flexible liposomes termed ethosomes. Liposomes were originally developed for parenteral administration of anticancer and antifungal agents. They are not as widely utilized for topical or transdermal drug delivery and are mainly included in cosmetics and long-acting sunscreen formulations, although their membrane structure is similar to that of natural membranes found in skin. Liposomes are expected to remain at the surface of the skin where they fuse with the skin lipids and slowly release the encapsulated drug [179,182,183,184].

Conversely, emulsions are heterogeneous, thermolabile biphasic liquids consisting of two or more immiscible fluids, which are made miscible through the addition of surfactants or emulsifying agents. They can be further categorized into micro- and nano-emulsions, as well as SEDDSs, depending on the formulation methods that are used. However, the instability of conventional emulsions causes difficulty in drug delivery [180,185]. Microemulsions technically are not emulsions, but are thermodynamically stable, single-phase systems that form spontaneously once the correct oil to water ratios are achieved. They possess a number of different microstructures depending on the nature and concentrations of the components. Alternatively, nanoemulsions are thermodynamically unstable dispersions that contain individual small droplets with a mean diameter of less than 200 nm. However, these dispersions possess reasonably good kinetic stability, as the droplets do not collide as frequently as in ordinary emulsions. The small droplet size furthermore allows for deep penetration into tissues as well as through fine capillaries. Positively charged (cationic) nanoemulsions increase skin permeation of poorly soluble drugs due to their interaction with the negatively charged skin epithelial cells. The main advantages of nanoemulsions include increased drug loading, enhanced drug solubility and bioavailability, reduced variability of drug uptake between different patients, controlled drug release, and protection from enzymatic degradation [178,180,182].

SEDDSs were originally developed from emulsions in order to generate a more physically stable formulation that is easier to prepare and can be used for oral drug delivery, as exemplified by their development specifically for oral administration of the immunosuppressant cyclosporine [178,185,186]. More recent activities focus on developing SEDDSs for other drug delivery routes of administration such as via the ocular-, vaginal- and sublingual routes [186,187,188,189,190]. Additionally, use of SEDDSs for treatment of dermal pathologies have been described [191,192]. SEDDSs are isotropic and thermodynamically stable emulsions containing an oil, surfactant, co-surfactant and drug(s) that spontaneously transform into oil-in-water emulsions upon the addition of aqueous media under gentle agitation [193]. SEDDSs are best suited for the incorporation of lipophilic drugs [194] as they are solubilized in the oily concentrate and then introduced into the aqueous media with mild agitation [178,185,186,194,195]. Interestingly, it has been demonstrated that SEDDSs encapsulating poorly water-soluble drugs elicited higher oral bioavailability in comparison with SEDDSs incorporating hydrophilic drugs; this clearly indicates the potential of SEDDSs to improve lipophilic drug delivery [196,197]. Nonetheless, for topical delivery of a drug such as clofazimine, the amount of clofazimine incorporated into the SEDDSs should be greater than the saturated concentration established in the oil phase in order to maximize clofazimine flux through the skin, as has also been noted for other poorly aqueous soluble drugs [197].

SEDDSs can be formulated in either nano- or micro droplets [198], where nanodroplet formation may occur in the presence of surface-active agents that cause spontaneous emulsification of finely dispersed droplets [194,199]. Formation of these finely dispersed nano-sized droplets follows from the intrinsic physicochemical properties of the constituents. Stability of these droplets depend upon the ratio of oil to surfactant [200]. However, it has been noted that high surfactant concentrations can cause tissue irritation [201]. Thus, spontaneous emulsification has to occur with decreased surfactant concentrations without reducing the stability of the formulations [202].

In view of the advantages of SEDDSs in terms of ease of formulation and solubilizing capacity for lipophilic drugs, SDEDDSs should provide even more benefits given the inclusion of multiple phases to optimize solubility and drug delivery via the dermal route of administration [203]. Drug delivery can be optimized by selecting immiscible oil phases in order to develop oil-in-oil-in water (O/O/W) or water-in-oil-in-oil (W/O/O) emulsions [203]. The employment of O/O/W systems will be beneficial if immiscible oil phases are used to create oil droplet formation in an inner oil phase within an outer oil phase [203]. This will provide a lipid matrix for enhancing solubility of lipophilic drugs while the water phase provides dermal hydration, which in turn establishes increased dermal drug delivery [204]. Conversely, W/O/O systems can provide a matrix for solubilization of both hydrophilic and lipophilic drugs with the added benefit of increased skin penetration as achieved by the outer oil phase that leads to enhanced dermal drug delivery [203,204,205].

As TB treatment requires combinations of different drugs administered over long periods of time as discussed above [37], the use of SDEDDSs can potentially provide an attractive dermal drug delivery vehicle by enabling simultaneous inclusion of several drugs within the hydrophilic and lipophilic phases of these multi-emulsions. Thereby, use of SDEDDSs represents a novel dermal drug delivery approach when compared to known dermal drug delivery vehicles. The lipid-based drug delivery systems discussed here are presented in Figure 5.

SDEDDSs are polydispersed systems (Figure 5) [206] that for oral drug delivery have the promise of increasing the oral bioavailability of drugs known to have low water solubility and high permeability, that is of the Biopharmaceutical classification system (BCS) Class II drugs. By transporting drugs dispersed in oil droplets while passing through the gastrointestinal tract (GIT) [206], they enable prolonged drug release within the GIT—the drugs incorporated in the inner-oil phase must partition across multiple phases before reaching the site of action where drug release is desired [206]. For topical application, the formulation in principle does not require prolonged release since the formulation is directly applied to the affected area to establish a localised therapeutic effect [207]. Nevertheless, the capacity for prolonged drug release provided by SDEDDSs can establish the intended therapeutic effect by controlling drug release as well as enabling the release of multiple drugs [206,207]. Conrolled drug release into the skin will obviate erratic drug release which can result in unwanted systemic uptake of the applied drug [207]. Use of conventional topical formulations that do not have controlled release properties place focus on establishing direct skin penetration of the encapsulated drugs [207]. In general, this is understandable as drugs must cross the protective barrier provided by the skin. This barrier reduces entry of drugs into the deeper skin layers thereby limiting dermal bioavailability which also compromises therapeutic effectiveness [207]. Therefore, optimal selection of excipients accompanied by a refined pre-formulation approach for preparing the SDEDDSs should overcome the challenges posed by drug solubilization and skin lipid disruption, and enable sustained release required to achieve dermal drug delivery [204,205,207].

SEDDSs and SDEDDSs additionally distribute drugs into the lymphatic system [178,180,196,208]. Due to absorption of SEDDSs and SDEDDSs through lymphatic vessels found in the dermis, drug delivery may be further enhanced [209,210]. Lymphatic absorption of clofazimine will provide an advantage as lymphadenitis is the most commonly observed manifestation of EPTB in both adults and children [68]. The lipid oil component is important for effecting lymphatic absorption through the dermis. Note that for oral administration, the oil has to solubilize a minimum of 50 mg/mL drug in order for lymphatic uptake to occur. In this respect, oils containing long chain triglycerides demonstrated increased lymphatic uptake, and the quantity of oil significantly contributes towards successful lymphatic uptake [171,186]. It was established that that a concentration of ≥25% oil content provides sufficient lymphatic absorption [131]. Unfortunately, lipids that are highly unsaturated may undergo autoxidation, and thus the addition of anti-oxidants must also be considered for topical drug delivery [208].

Not only is the evolution of SEDDSs moving towards alternative drug administration routes, but these systems are also being modified to advance muco-adhesion in order to extend contact time at the site of application [211,212,213]. Moreover, SEDDSs are being reformed to possess zeta-potential changing properties to improve drug delivery across mucus membranes [214]. Thus, combining mucolytic enzyme carrier systems with SEDDSs in order to enable local cleavability of mucus membranes resulted in increased drug permeation [215]. Consequently, SEDDSs have the potential to revolutionize drug delivery [187,194].

## 7. Conclusions

To summarize, the potential advantages of SEDDSs and SDEDDSs are enhanced drug entrapment capacity within physically more stable formulations, especially when compared to liposomes and emulsions. In addition, the formation of submicron droplets leads to an increase in the absorption surface area. SEDDSs elicit increased rates and amounts of drug absorption, thus increasing drug bioavailability. They moreover are able to deliver BCS Class II drugs efficiently. SEDDSs similarly have the potential for effective delivery of BCS class III drugs that display high solubility and decreased permeability properties, BCS class IV drugs that exhibit both low solubility and permeability characteristics, and hydrolytically susceptible drugs. Additionally, they offer consistent progressive diffusion profiles with reduced dosing and dosing frequencies. Moreover, upscaling techniques to prepare SEDDSs and SDEDDSs are simplified and cost effective compared to preparation of other lipid-based drug delivery systems.

Overall, the development of a topical SDEDDSs to aid in CTB treatment is potentially of considerable value as a reduced drug dose will be required to establish bioequivalence with oral drug delivery [201]. Due to the intractable nature of TB, high fixed doses of anti-tubercular drugs are normally recommended as inter-individual variability can lead to drug concentrations below the effective therapeutic dosage. Consequently, utilizing SDEDDSs for dermal administration of clofazimine will require decreased drug concentrations, will provide consistent drug delivery profiles that will be cytotoxic towards *Mtb* and thereby suppress drug resistance [216]. Importantly, such a route of administration should reduce the unpleasant discoloration associated with oral administration of clofazimine [87,210]. The improved drug loading capacity of SEDDSs can further support anti-tubercular treatment by either including higher concentrations of clofazimine or incorporating fixed-dose drug combinations with other antitubercular drugs [217,218]. In the latter respect, incorporation of both artemether and lumefantrine in a solid SEDDS double fixed dosage form was used successfully for treatment of malaria [217]. For TB, it should be possible to delay the development of anti-microbial resistance by co-administration of two drugs known to act synergistically [216]. For this reason, clofazimine and pyrazinamide, for example, can be combined into a SDEDDS formulation as synergism between the two drugs has already been established [71,219]. Moreover, recent synergism was reported from evaluating a combination of clofazimine, isoniazid and rifampicin against actively replicating planktonic organisms and slow replicating organisms [220]. Interestingly, first-line anti-tubercular drugs are not the only drugs that acts synergistically when used in combination with clofazimine. In a model study involving the use of macrophages infected with *Mtb*, synergism was detected between clofazimine and the novel prenylated amino-artemisinin WHN296 [221]. These significant findings support the contention that clofazimine will be a key component to most anti-tubercular drug regimens intended to treat active TB, resistant TB infections, and also CTB.

## Figures and Tables

**Figure 1 antibiotics-11-00806-f001:**
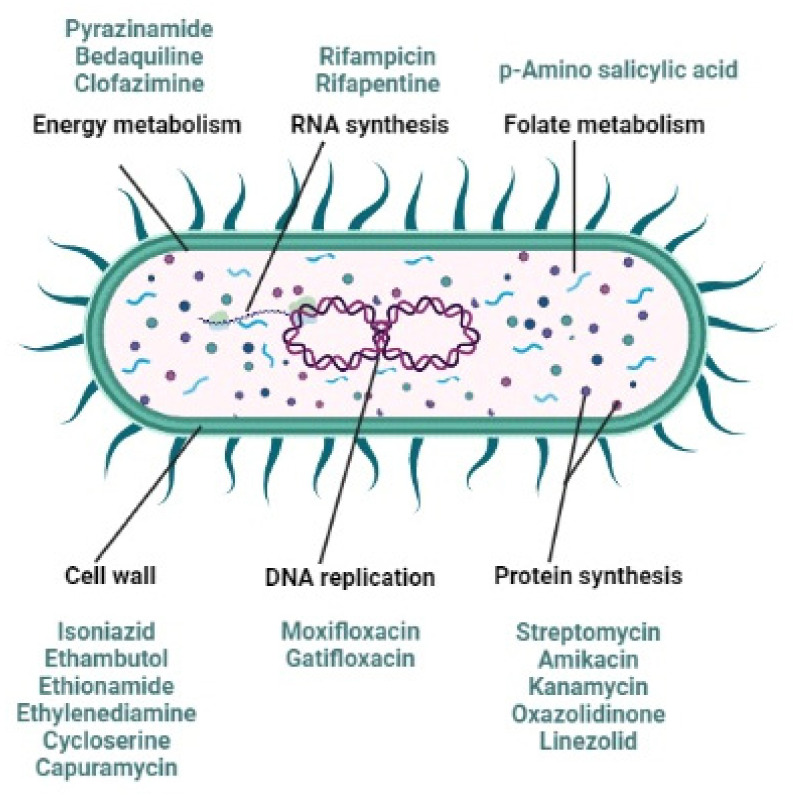
Mechanism of action of numerous anti-tubercular drugs (adapted from [47]).

**Figure 2 antibiotics-11-00806-f002:**
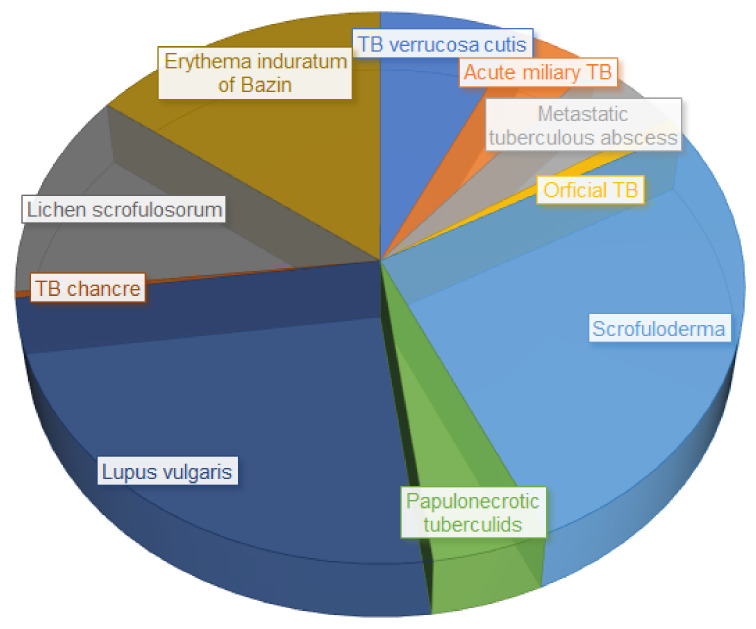
Incidence of different CTB manifestations (adapted from [75]).

**Figure 3 antibiotics-11-00806-f003:**
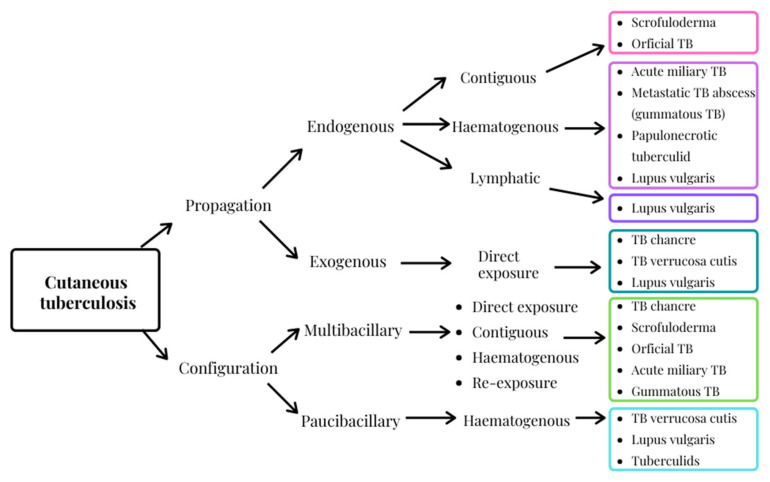
Classification system of CTB (adapted from [71,83]).

**Figure 4 antibiotics-11-00806-f004:**
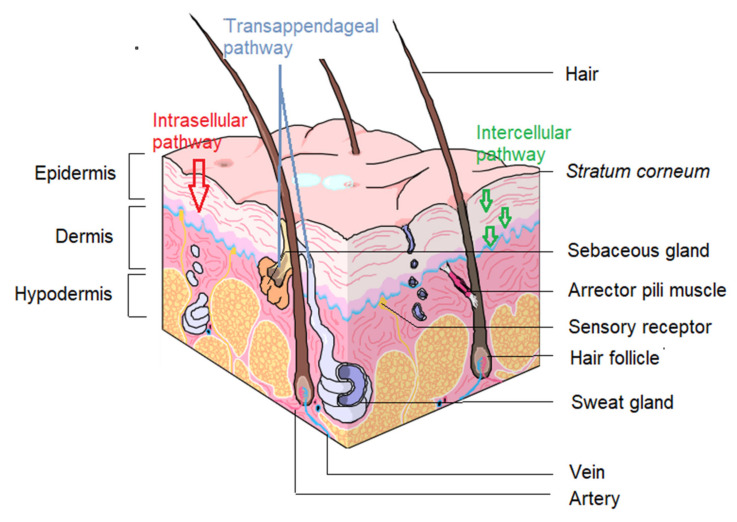
Dermal drug penetration pathways (adapted from [134]).

**Figure 5 antibiotics-11-00806-f005:**
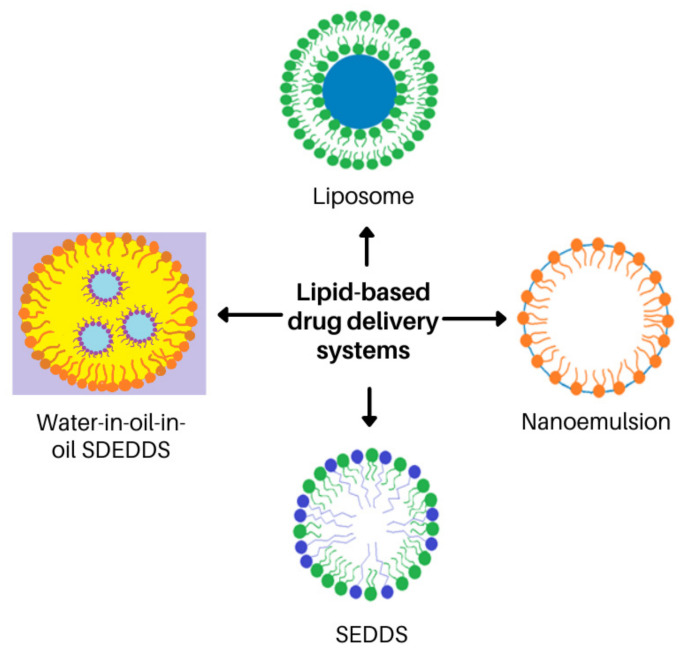
Lipid-based drug delivery systems [204,206].

**Table 1 antibiotics-11-00806-t001:** Recent MDR-TB drug classification system for promoting an extensive MDR-TB treatment regimen as established by the WHO, adapted from [51].

Priority Drug Groups	TB Drug
Group A: Incorporate all three drugs as part of regimen	Levofloxacin OR Moxifloxacin Bedaquiline Linezolid
Group B: Add one or both drugs to regimen	Clofazimine Cycloserine OR Terizidone
Group C: Add to complete regimen and include when drugs from Group A and B cannot be included as influenced by resistance, toxicity, or tolerability	Ethambutol Delamanid Pyrazinamide Imipenem–cilastatin OR Meropenem AND Amoxicillin/Clavulanate Amikacin OR Streptomycin Ethionamide OR Prothionamide Aminosalicylic acid

**Table 2 antibiotics-11-00806-t002:** Physicochemical Characteristics of Clofazimine ^a^.

Physicochemical Characteristics	
Chemical formula	C_27_H_22_Cl_2_N_4_
Molecular weight	473.40 D_a_
Chemical structure	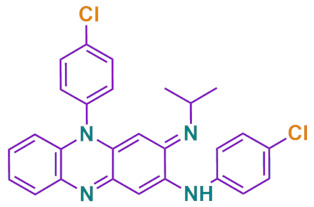
Melting point	210–212 °C
Appearance	Reddish-brown, fine powder
Log P(Octanol-water partition coefficient)	7.66
UV Detection wavelength	254 nm
Elimination half life	70 days
pK_a_ value	8.51
Solubility	Soluble in methylene chloride, very slightly soluble in ethanol, aqueous solubility < 0.001 mg/L

^a^ from [92,142,146].

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
