# Peer review of "Adapting Clofazimine for Treatment of Cutaneous Tuberculosis by Using Self-Double-Emulsifying Drug Delivery Systems"

_antibiotics, 2022, doi:10.3390/antibiotics11060806_

Round 1

Reviewer 1 Report

The review focused on the potential of utilizing self-double-emulsifying drug delivery systems for the treatment of cutaneous TB. To summary it, the authors introduced the current challenges for TB treatment, drug-resistant TB, the classification of TB and emphasized cutaneous TB. Then they summarized the current usage of Clofazimine and its limitation. It is meaningful to explore the alternative route of Clofazimine administration. It is helpful to better know the progress in the field. It covers the most relevant advances, historical literatures, and more recent progress. The references are related to the topic.

But the organization needs to be improved.

1.     line36 to line48 summarized TB incidence in developing and developed countries and raised drug-resistant TB. Then line49 to line58 introduced TB in other countries such as Australia, US and so on. At the end of this paragraph, it was said that children infection should be paid more attention. Following with this, line59 to line71 described the misdiagnosed or overlooked TB in children and also mentioned drug resistance in TB infected children. The authors aimed to introduce the current incidence of TB globally and more emphasis the challenges of TB treatment (drug resistance). However, the current writing cannot clearly reflect those aims, especially the challenges of TB treatment. The authors need to reorganize this part. In addition, introduction should include the purpose of the review using one or two sentences.

2.     Part 2 summarized anti-tubercular drug resistance in detail, which is good. line93-line100 showed MDR-TB. Then line101-113 introduced bacteria Mtb, active TB and latent TB. Logically, line101 to line113 should be before line93 to line100. In addition, line149 showed the definition of MDR-TB. Indeed, MDR-TB has appeared in line98. It is better to introduce the definition MDR-TB after the first appearance. The authors need to think about the logic flow clearly and reorganize it.

3.     The authors are talking about drug-resistant TB. However, this review only included the treatments for MDR-TB, but no summarized therapy for TB treatment. The authors should summarize the current treatment for TB, then TB resistance and then therapy for MDR-TB.

4.     This review seems to focus on cutaneous tuberculosis, according to the title and part 3. Line234 showed “the treatment for CTB is the same as systemic TB”. However, there is no introduction for systemic TB treatment. Clofazimine is not a drug specific to cutaneous tuberculosis, but for MDR-TB. Why did you describe Clofazimine to treat CTB? The manuscript should reflect it.

5.     There are some grammar problems. Some sentences are unreadable. Some words are not used appropriately. The authors should check the whole manuscript and improve the writing. Here didn’t list all of them.

            In line367 "An alternative means to deliver.....", it would be better to use "strategy" other than "means".

           Line45 to line47 is unreadable. The authors need to modify the sentence.

Reviewer 2 Report

The review article by Staden et al. 2022 entitled: Adapting Clofazimine for Treatment of Cutaneous Tuberculosis by Using Self-Double-Emulsifying Drug Delivery Systems s an interesting review that has brought a comprehensive understanding of the treatment of TB and its associated forms.

l will suggest that the authors add the chemical structures of these drugs. Since these structures could provide more visibility to the paper and increases its citation.
